# Flexible neural connectivity under constraints on total connection strength

**Gabriel Koch Ocker** [1] *, **Michael A. Buice** [1,2]

**1** Allen Institute for Brain Science, Seattle, Washington, United States of America, **2** Department of Applied Mathematics, University of Washington, Seattle, Washington, United States of America

\* gkocker@bu.edu

**Data Availability Statement:** The data used are available from the supplementary information of the publications describing them (as cited in the manuscript).

**Funding:** The author(s) received no specific funding for this work.

## Abstract

Neural computation is determined by neurons' dynamics and circuit connectivity. Uncertain and dynamic environments may require neural hardware to adapt to different computational tasks, each requiring different connectivity configurations. At the same time, connectivity is subject to a variety of constraints, placing limits on the possible computations a given neural circuit can perform. Here we examine the hypothesis that the organization of neural circuitry favors computational flexibility: that it makes many computational solutions available, given physiological constraints. From this hypothesis, we develop models of connectivity degree distributions based on constraints on a neuron's total synaptic weight. To test these models, we examine reconstructions of the mushroom bodies from the first instar larva and adult *Drosophila melanogaster*. We perform a Bayesian model comparison for two constraint models and a random wiring null model. Overall, we find that flexibility under a homeostatically fixed total synaptic weight describes Kenyon cell connectivity better than other models, suggesting a principle shaping the apparently random structure of Kenyon cell wiring. Furthermore, we find evidence that larval Kenyon cells are more flexible earlier in development, suggesting a mechanism whereby neural circuits begin as flexible systems that develop into specialized computational circuits.

## Author summary

High-throughput electron microscopic anatomical experiments have begun to yield detailed maps of neural circuit connectivity. Uncovering the principles that govern these circuit structures is a major challenge for systems neuroscience. Healthy neural circuits must be able to perform computational tasks while satisfying physiological constraints. Those constraints can restrict a neuron's possible connectivity, and thus potentially restrict its computation. Here we examine simple models of constraints on total synaptic weights, and calculate the number of circuit configurations they allow: a simple measure of their computational flexibility. We propose probabilistic models of connectivity that weight the number of synaptic partners according to computational flexibility under a constraint and test them using recent wiring diagrams from a learning center, the mushroom body, in the fly brain. We compare constraints that fix or bound a neuron's total

**Competing interests:** The authors have declared that no competing interests exist.

connection strength to a simple random wiring null model. Of these models, the fixed total connection strength matched the overall connectivity best in mushroom bodies from both larval and adult flies. We also provide evidence suggesting that neural circuits are more flexible in early stages of development and lose this flexibility as they grow towards specialized function.

## Introduction

The connectivity of neural circuits, together with their intrinsic dynamics, determines their computation. A goal of systems neuroscience is to uncover and describe these computational mechanisms in specific circuits. For example, associative memory is a quintessential neural computation [1–3]. In cerebellar and cerebral cortices, random connectivity may form high-dimensional representations to facilitate associative memory [4–7]. The synaptic weight distributions of Purkinje cells and cortical pyramidal neurons are consistent with optimal associative memory in simple models [8–13].

At the same time, neuronal connectivity is constrained by resource limitations and homeostatic requirements. The total strength of synaptic connections between two neurons is limited by the amount of receptor and neurotransmitter available and the size of their synapses [14]. Homeostatic synaptic scaling in pyramidal neurons of mammalian cortex and hippocampus regulates their total excitatory [15–18] and inhibitory [19–23] synaptic input strengths to regulate activity levels [24].

The neural connectivity necessary for a given computation must exist or develop within these physiological constraints. Furthermore, the computations performed by a circuit may require modification based on exposure to the environment and the needs of the organism. Physiological constraints could conflict with a required circuit configuration and pose a challenge to computational learning. It might thus be advantageous for circuits, under a fixed constraint, to enable a broad array of computations. We refer to connectivity patterns that can enable many computations as "flexible" under a constraint. Here we seek to understand whether this flexibility can predict neural connectivity patterns.

We examine the interaction between constraints and computations in the mushroom body, a cerebellum-like associative memory center in *Drosophila melanogaster* and other insects [25, 26]. Mushroom bodies are composed largely of Kenyon cells (KCs), which receive input from sensory projection neurons (PNs; Fig 1a). KCs also connect recurrently to each other, receive inhibitory and modulatory inputs, and project to mushroom body output neurons (MBONs). A combinatorial code for odorants in KCs [27–29] forms a substrate for associative learning at KC-MBON synapses [25, 30]. In *D. melanogaster*, neurons from a variety of circuits exhibit homeostatic regulation of connectivity during growth from the first instar larva to the third instar larva. This includes a homeostatic regulation of mechanosensory receptive fields [31], functional motor neuron outputs [32], and nociceptive projections [33]. Changes in inputs to central motor neurons elicit structural modifications to their dendrites that homeostatically maintain input levels [34]. Finally, changes in olfactory PN activity lead to homeostatic compensations in the number and size of their output synapses in the mushroom body [35–37]. Connections in the mushroom body also are generally limited to a few synapses per connection [38, 39]. We thus hypothesize that mushroom body connectivity in *D. melanogaster* might be structured to be computationally flexible under physiological constraints.

Consider the connections from one KC to its *K* targets. We will describe each projection by one synaptic weight, summed across synapses if the projection is multi-synaptic, so that this

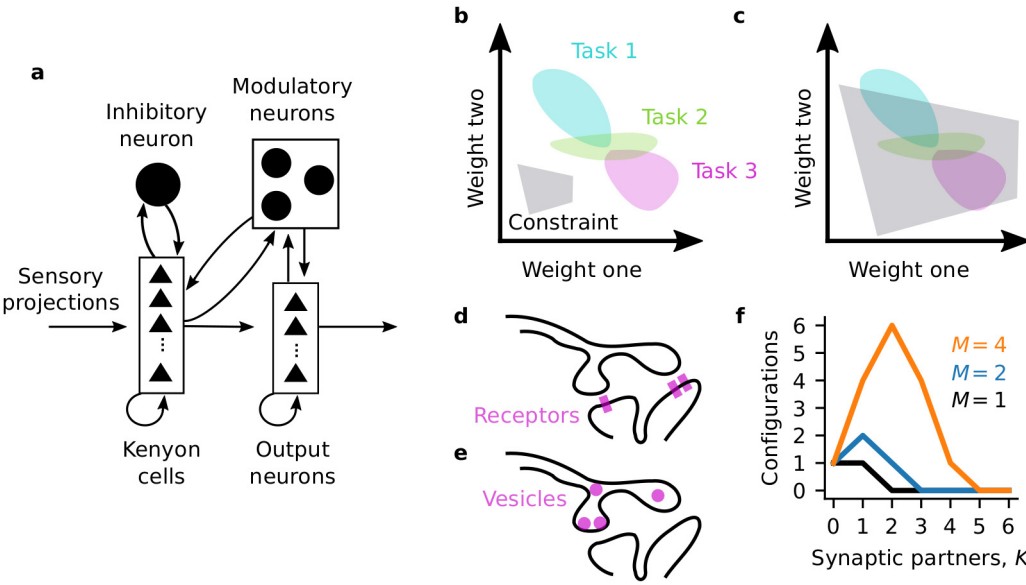

**Fig 1. Flexible connectivity under constraints.** (**a**) Mushroom body circuitry (cartoon based on [38]). (**b**) Synaptic weights occupy a *K*-dimensional space. *K* is the number of synaptic partners. The solution spaces for computational tasks are subspaces of the synaptic weight space, with dimension up to *K*. Constraints also define subspaces with dimension up to *K*. A tight constraint defines a small subspace with low potential overlap with computational solution spaces. (**c**) A loose constraint defines a large subspace, with greater potential overlap with computational solution spaces. (**d**) Cartoon of a postsynaptic resource constraint: a neuron with *M* = 3 units of postsynaptic weight (e.g., receptors) to distribute amongst two synaptic partners. (**e**) Cartoon of a presynaptic resource constraint a neuron with *M* = 4) units of synaptic weight (e.g., vesicles) to distribute amongst two partners. (**f**) Number of possible connectivity configurations for different values of *K* and *M* (given by the binomial coefficient).

KC's output weight configuration is a point in a *K*-dimensional synaptic weight space. Regions of this space might correspond to weight configurations that support different learned associations or computations (Fig 1b; "computation spaces"). A physiological or homeostatic constraint, such as those discussed above, also defines a region of allowed connectivity configurations (Fig 1b; "constraint space"). If a constraint is very tight, it might only allow a few configurations, and even disallow computationally useful configurations as in Fig 1b, where the constraint and computation spaces do not overlap. In order for the KC to be both healthy and computationally useful, its connectivity must lie in the intersection of the constraint space and a computation space. This is easier if the constraint space is large—if the KC's connectivity is flexible under its constraints. (Fig 1c). "Flexibility" thus refers to the number of possible joint configurations of all this KC's outputs and is a property of the neuron's full *K*-dimensional output connectivity rather than of an individual synapse. While we discussed the flexibility of a KC's output connectivity here, the same idea can be applied to its inputs, or those of MBONs, or of other neurons in other systems.

In this study, we formulate this idea for simple models of two constraints: a (1) bounded or (2) fixed total connection strength. We propose that circuits might face a pressure to be flexible under these constraints. This motivates probabilistic models for the number of synaptic partners to a neuron. We test these models against each other and a simple random wiring null model using recently available electron microscopic wiring diagrams of the mushroom body from larval [38] and adult [39] *D. melanogaster*. We found that overall, the fixed net weight model provided the best description for neurons' numbers of synaptic partners in the mushroom bodies. The one exception we saw was in the most mature KCs of the larval mushroom body, which were better described by a binomial random wiring model. This suggests a

developmental progression in the pressures shaping KC wiring in the first instar larva of *D. melanogaster*.

## Results

### Measuring constraint flexibility

We begin with a simple example where a neuron has *M* units of synaptic weight, of size Δ*J*, available. These could correspond, for example, to individual receptors or vesicles. The neuron can assign these synaptic weight units to its *K* partners (presynaptic partners for receptors, Fig 1d, or postsynaptic partners for vesicles, Fig 1e). We will also call the number of synaptic partners the degree or connectivity degree.

To measure how flexible the neuron is with *M* synaptic weight units and *K* partners, we can count possible connectivity configurations. Since the constraint treats all synaptic partners symmetrically, the number of possible configurations is given by the binomial coefficient "M choose K". For *M* = 4 and degree two, there are six possible configurations. With *M* = 4 and degree three, there are four possible configurations. Thus with the constraint of *M* = 4, the neuron is more flexible with two connections than three since there are more ways to satisfy the constraint. The neuron's flexibility under this constraint (of fixed total synaptic resources *M*Δ*J*) is a function only of the number of synaptic partners *K* only. For different numbers of synaptic weight units *M*, the flexibility exhibits different profiles as a function of the degree *K* (Fig 1f).

Synaptic weights can be made up of many small units of strength, corresponding to (for example) individual receptors or vesicles. So, we will model individual synaptic weights as continuous variables rather than the discrete description above. Throughout, we will use "synaptic weight" and "connection strength" interchangeably to refer to the total strength of projections from one neuron to another.

### Degree distributions from constraints on net synaptic weights

We consider a simple model of synaptic interactions where a neuron has *K* synaptic partners and the strength of projection *i* is $J_i$. The first constraint we consider is an upper bound on the total connection strength:

$$\sum_{i=1}^{K} J_i \leq \bar{J} \tag{1}$$

The bound $\bar{J}$ could be interpreted multiple ways, for example as a presynaptic limit due to the number of vesicles currently available before more are manufactured or a postsynaptic limit due to the amount of dendritic tree available for synaptic inputs. The value of $\bar{J}$ could be set by metabolic or resource constraints. Rather than modeling the biological origin of $\bar{J}$, we will focus on the structure this constraint imposes in the *K*-dimensional synaptic weight configuration space.

With *K* synaptic partners, the constraint (Eq 1) defines a *K*-dimensional volume. For a neuron with two synaptic partners, this is the portion of the plane bounded by the axes and a line that stretches from $(0, \bar{J})$ to $(\bar{J}, 0)$ (Fig 2a). For three synaptic partners, the weight configurations live in three-dimensional space and are constrained to lie in the volume under an equilateral triangle (Fig 2b). It is equilateral because its vertices are defined by the configurations where one connection uses the total weight, $\bar{J}$. In general, for *K* synaptic partners the synaptic weights live in the volume under a *K* − 1 dimensional simplex (the geometric generalization of

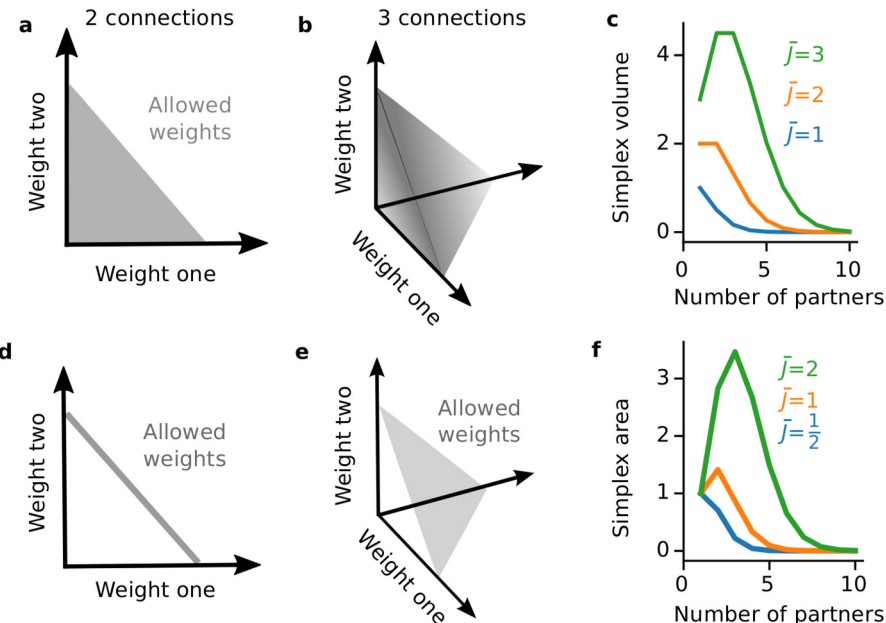

**Fig 2. Constraints on total synaptic weights. (a-c)** An upper bound on the total synaptic weight. **(d-f)** A fixed total synaptic weight. **(a)** For two inputs with a total synaptic weight of at most $\bar{J}$, the synaptic weights must live in the area under a line segment from $(0, \bar{J})$ to $(\bar{J}, 0)$ (a regular 1-simplex). **(b)** For three inputs, the synaptic weights must live in the volume under a regular two-simplex. **(c)** Volume of the $K-1$ simplex as a function of the number of presynaptic partners, $K$, for different maximal net weights $\bar{J}$. **(d)** For two inputs with a total synaptic weight fixed at $\bar{J}$, the synaptic weight configurations must be on the line segment from $(0, \bar{J})$ to $(\bar{J}, 0)$. **(e)** The solution space for the fixed net weight constraint with three inputs is an equilateral planar triangle (a regular 2-simplex). **(f)** Surface area of the regular $K-1$ simplex as a function of the number of presynaptic partners, $K$, for different net synaptic weights $\bar{J}$.

a triangle to higher dimensions). This $K$-dimensional volume is [40]

$$V(K, \bar{J}) = \frac{\bar{J}^K}{K!} \qquad (2)$$

This extends the counting model of Fig 1d–1f to the case of a large pool of synaptic resources (Measuring synaptic weight configuration spaces) and measures the size of the space of allowed circuit configurations under the constraint of Eq 1. The volume under the simplex increases with the max weight $\bar{J}$, but for $\bar{J} \geq 2$ it has a maximum at $K \geq 1$ and then decays (Fig 2c).

Under our hypothesis of flexible computation under a constraint, there should be a pressure towards circuit structures with a large number of allowed synaptic weight configurations. There may be other competing pressures on the circuit architecture, which prevent it from being optimally flexible (achieving the largest number of possible configurations). To model the pressure towards flexibility, we thus stipulate that the probability of having $K$ synaptic partners given resource limits $\bar{J}$ is proportional to the number of possible configurations, i.e., the volume of the weight space with $K$ partners. For a bounded net synaptic weight:

$$p_V(K|\bar{J}) = \frac{V(K, \bar{J})}{Z_V(\bar{J})} \qquad (3)$$

where the subscript $V$ marks the probability distribution as proportional to the simplex's

volume and the normalization constant, $Z_V$, ensures that the probability distribution sums to 1:

$$Z_V(\bar{J}) = \sum_{K=1}^{\infty} V(K, \bar{J}) \tag{4}$$

This normalization constant can be computed exactly to reveal a zero-truncated Poisson distribution:

$$p_V(K|\bar{J}) = \frac{\bar{J}^K}{(\exp(\bar{J}) - 1)K!} \tag{5}$$

Note that this is a distribution for the number of synaptic partners to a neuron (its degree), not for its synaptic weights. The degree distribution is conditioned on the maximum total synaptic weight, as measured by the parameter $\bar{J}$.

In addition to facing resource constraints, neurons also homeostatically regulate their total input strengths. Motivated by this, the next constraint we consider holds the total synaptic weight fixed at $\bar{J}$:

$$\sum_{i=1}^{K} J_i = \bar{J} \tag{6}$$

This constraint is satisfied on the surfaces of the same simplices discussed above (Fig 2d and 2e). Their surface area is (Measuring synaptic weight configuration spaces):

$$A(K, \bar{J}) = \frac{\bar{J}^{K-1}\sqrt{K}}{(K-1)!} \tag{7}$$

Like the simplex's volume, the surface area increases with the total synaptic weight $\bar{J}$ but for $\bar{J} \gtrsim 0.7$ it has a maximum at $K \geq 1$ (Fig 2f). We will also examine the size of this constraint as a model for degree distributions:

$$p_A(K|\bar{J}) = \frac{A(K, \bar{J})}{Z_A(\bar{J})} \tag{8}$$

where $p_A$ denotes the probability proportional to the simplex's surface area and the normalization constant is

$$Z_A(\bar{J}) = \sum_{K=1}^{\infty} \frac{\bar{J}^{K-1}\sqrt{K}}{(K-1)!} \tag{9}$$

In contrast to the bounded net weight model (Eq 5), we are not aware of an exact solution for $Z_A$. When required, we will either approximate it by truncating at large $K$ or bound it (Model comparison: Fixed net weight model).

## Testing degree distribution models

To test these models, Eqs 5 and 8, requires joint measurements of neurons' total synaptic weight and number of synaptic partners. One type of data with measurements reflecting both of these are dense electron microscopic (EM) reconstructions with synaptic resolution, where (in a large enough tissue sample) all of a neuron's synaptic partners can be identified and the size or number of synapses provide indirect measurements of the connection strength.

The published EM wiring diagrams of *D. melanogaster* mushroom bodies measure synaptic strengths by the count of synapses [38, 39]. While we are not aware of joint measurements of synapse counts and physiological connection strength in Kenyon cells, the relationship of anatomical and physiological measures of connection strength has been studied in mammalian pyramidal neurons. There, synapse size and synapse strength are highly correlated [41–45]. We thus assumed that the total number of synapses onto a neuron, $\bar{S}$, is proportional to its total connection strength constraint:

$$\bar{J} = \alpha \bar{S} \tag{10}$$

where $\alpha$ is the unknown constant relating the synapse count and the net synaptic weight. This assumption introduces $\alpha$ as an additional parameter in our degree distribution models, Eqs 5 and 8, so that each neuron's degree is conditioned on two things: the unknown parameter $\alpha$ and that neuron's number of synapses $\bar{S}$.

In addition to the two constraint-inspired models of Eqs 5 and 8, we also examined a simple random wiring null model where the number of partners follows a zero-truncated binomial distribution:

$$p_B(K|N, q) = \binom{N}{K} \frac{q^K (1 - q)^{N-K}}{1 - (1 - q)^N} \tag{11}$$

This binomial wiring model assumes that each of $N$ potential synaptic partners to a Kenyon cell has a fixed probability $q$ of making a connection, and that whether or not different potential partners actually connect is independent. We used anatomical measurements for $N$ (Model comparison: Zero-truncated binomial model) and took $q$ as an unknown parameter for this model. Note that this is a binomial model for connections, in contrast to the binomial model for resource allocation of Fig 1d–1f.

To measure and compare how well these models explain KC connectivity we computed their Bayesian evidence: the likelihood of the data under a model, marginalizing over the unknown parameters (Model comparison). For the fixed net weight, for example, the evidence is

$$p_A(K|\bar{S}) = \int_0^\infty d\alpha \, p(\alpha) \prod_i p_A(K_i | \alpha, \bar{S}_i) \tag{12}$$

where $i$ indexes KCs and $p(\alpha)$ is a prior distribution for $\alpha$. We have a corresponding integration over $\alpha$ to find the model evidence of the bounded net weight model, and for the binomial wiring model an integration over the unknown connection probability $q$.

Calculating the model evidence requires choosing a prior distribution for the unknown parameter ($p(\alpha)$ in Eq 12). We will use flat priors, as well as the Poisson Jeffreys prior. (Jeffreys priors maintain uncertainty under different scaling or unit choices.) The normalization constant for the fixed net weight model ($Z_A$, Eq 9) was not analytically tractable, so we computed upper and lower bounds for it. These bounds for $Z_A$ then gave us bounds on the fixed net weight model's evidence (Model comparison: Fixed net weight model).

## Larval Kenyon cell outputs

We first examined KCs outputs in the first instar larva, using the complete synaptic wiring diagram of its 223 KCs (110 on the left side of the brain and 113 on the right) from [38]. We excluded projections to the inhibitory APL neuron and the modulatory dopaminergic, and octopaminergic neurons as well as interneurons so the out-degree of each KC measures its

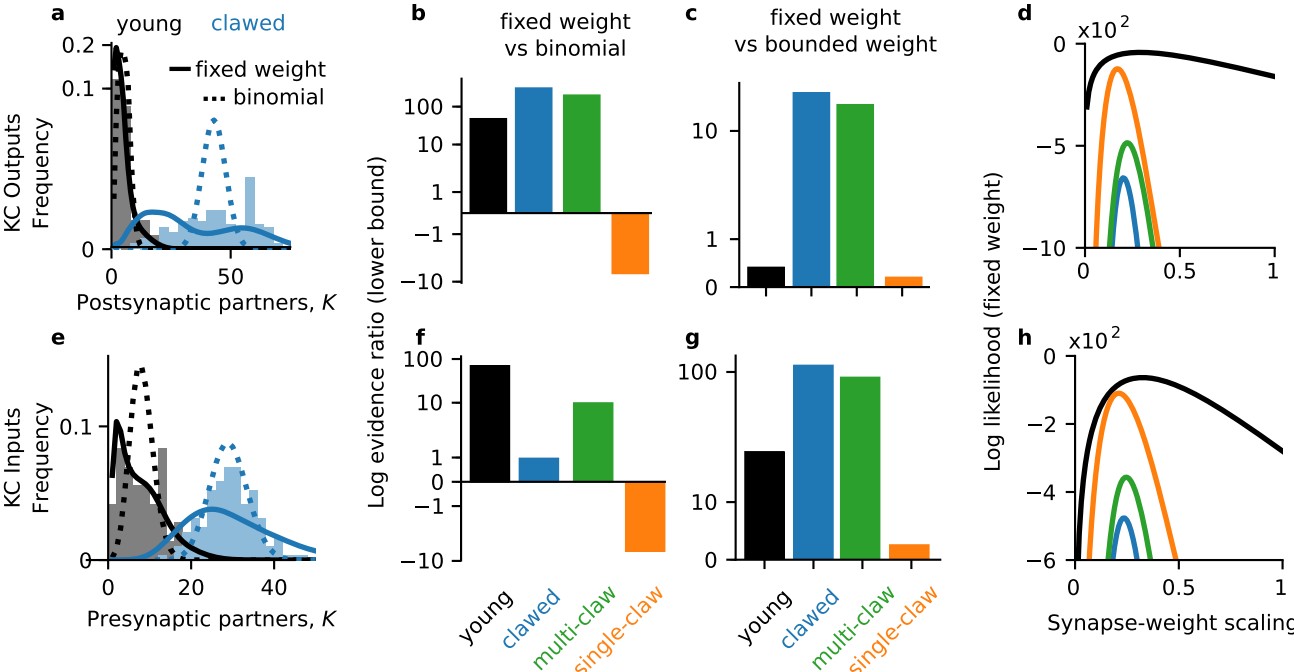

**Fig 3. Kenyon cell degree distributions in larval *D. melanogaster*.** (**a**) Distribution of number of postsynaptic partners for larval KCs. Shaded histograms: empirical distribution. Solid lines: the marginal simplex area distribution at the maximum likelihood value of $\alpha$, after integrating out the number of synapses $\bar{S}$ against its empirical distribution. Dotted lines: the maximum likelihood binomial distribution. (**b**) Lower bound for the log evidence ratio (log odds) for the fixed net weight and binomial wiring models. Positive numbers favor the fixed weight model. Evidences computed by a Laplace approximation of the marginalization over the parameters. (**c**) Lower bound for the log odds for the fixed net weight and bounded net weight models; positive numbers favor the fixed weight model. (**d**) Log likelihood of the fixed net weight model ($\sum_i \ln p_A(K_i \mid \bar{S}_i, \alpha)$) as a function of the scaling between synapse counts and net synaptic weights, $\alpha$. (**e-h**) Same as (**a-d**) but for inputs to larval KCs.

number of postsynaptic KCs and MBONs. (There are 48 MBONs, 24 on each side of the brain.) We obtained similar results as reported here, however, when including those other synapses. As in previous studies, we used only reliable multi-synapse connections (but obtained similar results as reported when including single-synapse connections) [38, 39]. Larval KCs can be morphologically classified by their age. The dendrites of mature KCs form claws around PN axons. The 78 young KCs do not have claws, and the 36 single-claw KCs are older than the 109 multi-claw KCs [38]. KCs have a wide range of presynaptic degrees, with very different out-degree distributions for young and clawed KCs (Fig 3a histograms).

As a first test, we examined the maximum likelihood marginal degree distributions. That is, we computed the maximum likelihood value of $\alpha$ to obtain the conditional degree distribution $p(K \mid \hat{\alpha}, \bar{S})$ and then marginalized out the number of synapses, $\bar{S}$, using its observed distribution to compute $p(K|\hat{\alpha})$ (Fig 3a, solid lines). For the binomial model we computed the maximum likelihood value of the connection probability $q$ (Fig 3a, dashed lines). For the young and clawed KCs, the fixed net weight model appeared to match the marginal degree distributions better than the binomial wiring model (Fig 3a).

To quantitatively compare how two models explained the data, we computed their log evidence ratio (log odds). The log odds for the fixed net weight model $p_A$ versus the bounded net weight model $p_V$ are

$$L = \ln \frac{p_A(K|\bar{S})}{p_V(K|\bar{S})} \tag{13}$$

and positive $L$ favors the fixed net weight model, while negative $L$ favors the bounded net weight model. For example, $L = 1$ means that the data are $\exp(1) \approx 2.72$ times more likely under the fixed net weight model than the bounded net weight model and $L = 10$ means the data are $\exp(10) \approx 22026.47$ times more likely under the fixed net weight model.

We found that the log odds favored the fixed net weight model over the binomial wiring model for young and multi-claw KCs (Fig 3b; log odds at least 48.17 and 201.8 respectively), but not for single- or two-claw KCs (Figure Ab in S1 Figs; log odds at most -0.78 and -0.50 respectively). For KCs with three or more claws, the log odds for the fixed weight model over the binomial wiring model were at least 20.3 (Figure Ab in S1 Figs). The log odds favored the bounded net weight model over the binomial wiring model in the same cases: for young and clawed KCs, except for single- and two-claw KCs (Figure Ac in S1 Figs). The fixed net weight model described KC output degree distributions better than the bounded net weight for all types of KC in the larva (Fig 3c, Figure Aa in S1 Figs). To control for our choice of prior (e.g., $p(\alpha)$ in Eq 12) we also performed a model comparison using the Jeffreys prior for the Poisson distribution (Model comparison) for $\alpha$ and $q$. Under the Poisson Jeffreys prior, the only result that changed was that the bounded net weight was the best model for the young KC outputs (Figure Ad-f in S1 Figs).

We next asked whether the relationship between anatomical and physiological synaptic weights exhibited a similar developmental trajectory as the model likelihoods, with multi-claw KCs appearing more similar to young KCs. To this end, we examined the likelihood of the data under the fixed net weight model as a function of the scaling parameter $\alpha$ (Fig 3d). The maximum likelihood values of $\alpha$ decreased with KC age, from 0.29 for young to 0.22 for multi-claw and 0.17 for single-claw KCs (Fig 3d). The two-claw KCs exhibited a similar scaling as the single-claw KCs (0.17), while KCs with three or more claws had higher $\alpha$ values (0.24 for three- and four-claw KCs, 0.28 for five- and six-claw KCs). This suggests that the relationship between net synapse counts and regulated net synaptic weights in the larval mushroom body may become weaker during KC maturation.

## Larval Kenyon cell inputs

We next examined the inputs to KCs in the larva. Like for the outputs, we examined multi-synapse connections and excluded inputs from the inhibitory APL and modulatory neurons (but obtained similar results when including them). There is a wide distribution of in-degrees for both larval and clawed KCs (Fig 3e, histograms). The maximum likelihood fit of the fixed net weight model appeared a much better fit than the maximum likelihood binomial for young KC inputs (Fig 3e, black solid vs dashed curves). For clawed KCs, it was less immediately clear which maximum likelihood model better explained the in-degree distribution (Fig 3e, blue solid vs dashed curves).

We again examined the log odds for pairs of models. The log odds favored the simplex area model over the binomial model for both young and clawed KCs (Fig 3f, log odds at least 70.46 and 0.97). We next asked whether this depended on the number of claws. The log odds favored the fixed net weight model over the binomial wiring model for multi-claw KCs, but not single-claw KCs (Fig 3f, log odds at least 9.92 for multi-claw KCs; Figure Bb in S1 Figs, log odds at most -1.57 for single-claw KCs). Within multi-claw KCs, the log odds also favored the binomial model over the fixed weight model for two-claw KCs, but not for KCs with at least three claws (Figure Bb in S1 Figs). We found similar results comparing the bounded net weight model with binomial wiring (Figure Bc in S1 Figs). The log odds favored the fixed net weight model over the bounded weight model for all KC types (Fig 3g). We obtained similar results under the Poisson Jeffreys prior for $\alpha$ and $q$ (Figure B in S1 Figs). Since single-claw KCs are

more mature than multi-claw KCs [38], these results together suggest that flexibility under a homeostatically fixed net weight governs KC input connectivity early in development, with other factors shaping connectivity after sensory and behavioral experience.

We next asked whether the relationship between anatomical and physiological input synaptic weights exhibited a similar developmental trajectory. We saw that that the maximum likelihood value of $\alpha$, $\hat{\alpha}$, decreased with KC age (Fig 3h); maximum likelihood values of $\alpha$: 0.33, 0.25, and 0.21 for young, multi-claw, and single-claw KCs, respectively). The scaling for single- and two-claw KCs were similar ($\hat{\alpha}$ of 0.2 for two-claw KCs), while KCs with three or more claws had $\hat{\alpha} \geq 0.25$. Under the simple model of Eq 10, these suggest that the translation of synapse counts into a physiologically regulated net synaptic weight becomes weaker during KC maturation. This could relate to the spatial concentration of synapses in claws of the dendrite.

## Larval MBON inputs

The next stage of mushroom body processing after KCs occurs at MBONs (Fig 1a). They exhibit a wide range of in-degrees (Fig 4a) from three presynaptic KCs (for the left MBON-n1) to 105 presynaptic KCs (for the left MBON-m1) [38]. Neither of the max likelihood fixed net weight (Fig 4a, black) and binomial (Fig 4a, blue) models appeared to be as good fits for the MBON in-degree distribution as they were for the KCs (Fig 3a and 3d). The fixed net weight model matched the breadth of the degree distribution, however, while the binomial model did not. We observed similar results for the bounded net weight model. To test which model provided a better explanation of the data overall, not just at a single parameter value, we again computed their log odds (Fig 4b). The log odds favored the fixed net weight model over both the bounded net weight (log odds at least 7.70) and binomial models (log odds at least 249.44). This was despite the fixed and bounded net weight models' likelihoods being sharper functions of $\alpha$ than the binomial model's likelihood as a function of $q$ (Fig 4c).

In summary, we found that the degree distribution predicted by flexible wiring under a homeostatically fixed total connection strength was the best overall model for KC input and output degrees, and MBON input degrees (Figs 3 and 4). The one exception to this were the single-claw KCs, which were best described by a binomial wiring model (Fig 3b and 3f).

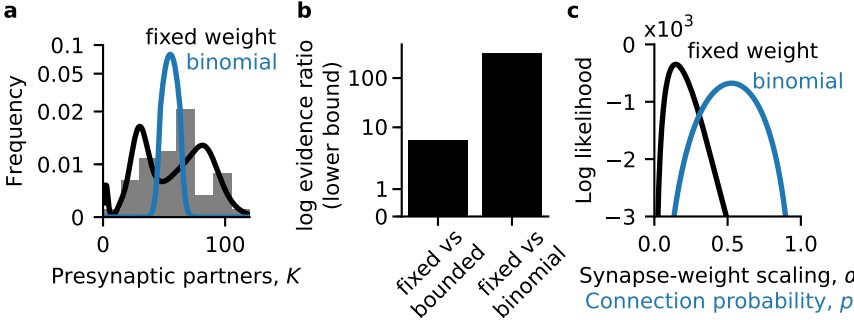

**Fig 4. Mushroom body output neuron degree distributions in larval *D. melanogaster*.** (**a**) Number of inputs to MBONs. Shaded histograms: empirical distribution. Black curve: the marginal simplex area distribution at the maximum likelihood value of $\alpha$, after integrating out the number of synapses $\bar{S}$ against its empirical distribution. Blue curve: the maximum likelihood binomial distribution. (**b**) Lower bound for the log evidence ratio (log odds) for the fixed net weight and binomial wiring models. Positive numbers favor the fixed weight model. Evidences computed by a Laplace approximation of the marginalization over the parameters. (**c**) Likelihood vs model parameter for the fixed net weight (black) and binomial (blue) models.

## Adult Kenyon cell outputs

To test the generality of these results, we turned to a related circuit: the adult *D. melanogaster* mushroom body. It contains the same general types of cells as the larva, though in different numbers, with the same broad circuit structure (Fig 1a). We examined a recent connectome of the alpha lobe of the adult mushroom body from Takemura et al. [39]. The alpha lobe is defined by KC axons, so these data do not include the PN inputs which target dendrites. It contains the axons of 132 alpha prime lobe KCs and 949 alpha lobe KCs. Like in the larva, the age of adult KCs can be classified morphologically. KCs of the alpha prime lobe are born before KCs of the alpha lobe. In the alpha lobe, the 78 posterior KCs are born before the 480 surface KCs, which are in turn born before the 259 core KCs [46, 47].

Since the adult data are only for axo-axonal connectivity, we first repeated our previous analysis without KC-MBON connections to examine the axo-axonal KC output connectivity in the larva. We found similar results as for the full connectivity (Figure Cf, g in S1 Figs).

In the adult, Kenyon cells had heterogenous out-degrees, with alpha lobe KCs exhibiting a bimodal distribution (Fig 5a). This bimodality was reflected in the out-degrees of posterior, core and surface KCs, rather than arising from the separate alpha lobe KC types. The fixed net weight models predicted adult KC out-degree distributions better than the binomial wiring model for all KC types (Fig 5b), as did the bounded net weight model (Figure Df, in S1 Figs). The fixed net weight provided a better description for all the out-degree distributions of all types of adult KC than the bounded net weight except when posterior, core, and surface KCs were all considered together (Fig 5c blue). These results were consistent when using the Poisson Jeffreys prior for $\alpha$ and $q$ (Figure Dd-f in S1 Figs). In summary, the degree distributions of

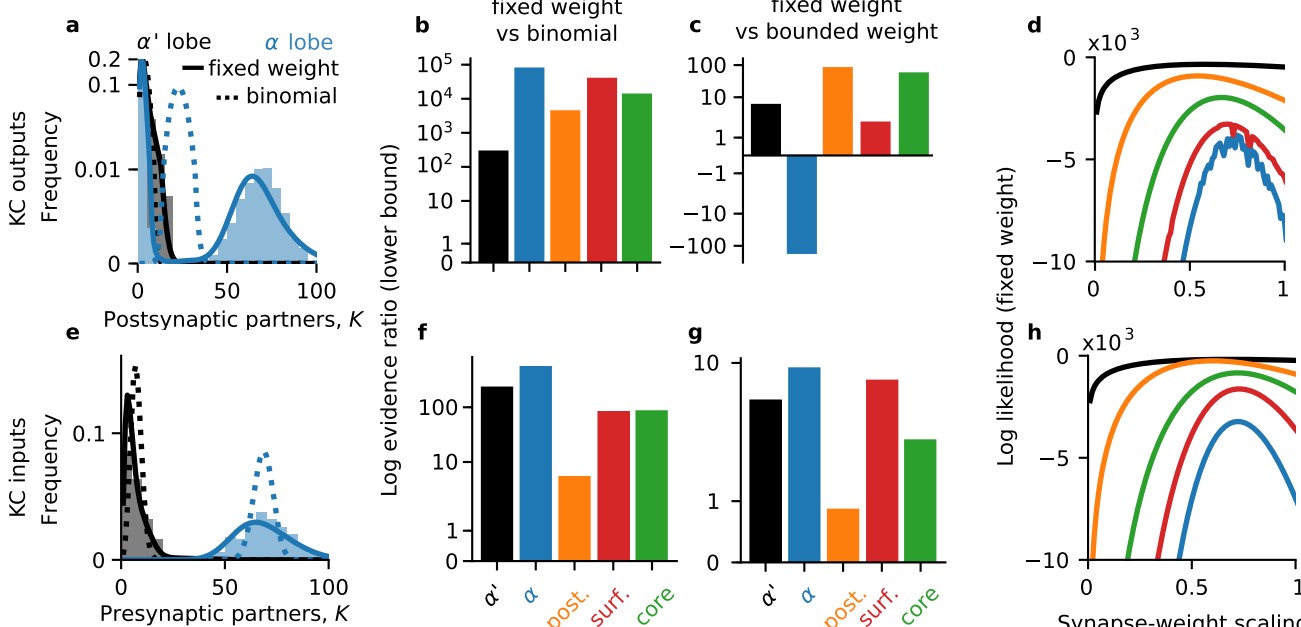

**Fig 5. Kenyon cell degree distributions in adult *D. melanogaster*.** (**a**) Distribution of number of postsynaptic partners for adult KCs. Shaded histograms: empirical distribution. Solid lines: the marginal simplex area distribution at the maximum likelihood value of $\alpha$, after integrating out the number of synapses $\bar{S}$ against its empirical distribution. Dotted lines: the maximum likelihood binomial distribution. (**b**) Lower bound for the log evidence ratio (log odds) for the fixed net weight and binomial wiring models. Positive numbers favor the fixed weight model. Evidences computed by a Laplace approximation of the marginalization over the parameters. (**c**) Lower bound for the log odds for the fixed net weight and bounded net weight models; positive numbers favor the fixed weight model. (**d**) Log likelihood of the fixed net weight model as a function of the scaling between synapse counts and net synaptic weights. (**e-h**) Same as (**a-d**) but for inputs to adult KCs.

KC outputs in the adult alpha lobe are best described by flexibility under a homeostatically fixed net synaptic weight.

The scaling between synapse counts and synaptic weights varied by KC type in the adult (Fig 5d). The maximum likelihood values of $\alpha$ were 0.58, 0.54, 0.7 and 0.67 for alpha prime, posterior, surface and core KCs respectively (in approximate developmental order). These suggest a divide with more mature KCs having more synaptic weight per synapse, on average, than younger KCs. The log likelihood for the alpha prime KCs was also much less sensitive to low values of $\alpha$ than the other KC types (Fig 5d black vs colored curves), suggesting a more heterogenous or flexible relationship between axonal input synapse counts and regulated output synaptic weights similar to for the input synaptic weights (Fig 5d black vs colored curves).

### Adult Kenyon cell inputs

As in the larva, adult KCs exhibited a range of in-degrees. KCs in the alpha prime lobe receive fewer axonal inputs than KCs in the alpha lobe (Fig 5e). As before, we computed the maximum likelihood marginal degree distributions, and saw that the binomial model appeared much worse than the fixed net weight model (Fig 3e solid vs dashed lines). This observation was born out by the models' evidences.

For comparison with the adult data, we again examined KC-KC connectivity in the larva, neglecting the inputs from projection neurons onto KC dendrites, and found similar results as for the full connectivity (Figure Ca-d in S1 Figs).

In the adult, the fixed net model explained the in-degree distribution of every KC type better than the binomial model (Fig 5f; log odds at least 232.57, 551.5, 5.36, 85.82, and 83.02 for alpha prime, all alpha lobe, posterior, core and surface KCs respectively). The fixed net weight model also explained the in-degree distributions better than the bounded weight model (Fig 5f; log odds at least 4.59, 9.0, 0.87, 2.0, and 6.96 for alpha prime, all alpha lobe, posterior, core and surface KCs respectively). The upper bounds for the log odds of the fixed net weight distribution were close to the lower bounds for the adult KCs (Figure Ea, b in S1 Figs). We found similar results using the Poisson Jeffreys prior for the unknown parameters $\alpha$ and $q$ (Figure Ed-f in S1 Figs). Together with the consistent results for adult KC output degrees (Fig 3e–3g), these suggest that flexibility under a fixed net synaptic weight governs KC connectivity in the alpha lobe of the mushroom body.

The scaling between the synapse count and net synaptic weight, $\alpha$, exhibited similar patterns for adult KC inputs and outputs (Fig 5d vs 5h). The maximum likelihood values for $\alpha$ were 0.68, 0.6, 0.73, and 0.72 for alpha prime, posterior, surface, and core KCs (ordered from approximately oldest to youngest). These suggest that in the alpha lobe, surface and core KCs have more input synaptic weight per synapse on average than posterior KCs. The log odds for the alpha prime KCs was much less sensitive to small values of $\alpha$ than the classes of alpha lobe KCs (Fig 5h, black vs colored curves), suggesting a more heterogenous or flexible relationship between axonal input synapse counts and regulated synaptic weights in alpha prime KCs.

### Measuring the cost of changing joint synaptic weight configurations

In measuring the flexibility of connectivity under a constraint, we measured the difference between two synaptic weight configurations by their straight-line (Euclidean) distance: the root sum squared difference in each synaptic weight. This was the origin of $\sqrt{K}$ in the surface area of the simplex (Eq (7)). It corresponds to the assumption that different connections can potentiate or depress simultaneously: for example, that a vesicle can be taken from one connection, depressing it, and given to another connection to potentiate it (Fig 1e). This implies

that the cost of changing one synaptic weight by an amount $d$ is the same as that of changing two weights by $d/\sqrt{2}$ each.

Potentiating one connection and depressing another might, however, have separate costs. This can be modeled by choosing a different norm for the space of synaptic weight configurations. For example, a neuron's connections might potentiate or depress separately so the cost of changing one connection by $d$ is the same as the cost of changing two connections by $d/2$. In this case distances between configurations are measured by the 1-norm given by the sum of absolute differences. This changes the measure of the surface area of the simplex, replacing $\sqrt{K}$ by $K$ in Eq 7. This does not change the results of our analysis of KC input connectivity (Figures F, G in S1 Figs). For KC output connectivity, the bounded net weight was a better model than the fixed net weight, under the 1-norm for weight changes, for larval young KCs (Figure H in S1 Figs) and adult surface KCs (Figure I in S1 Figs).

## Optimally flexible connectivity

Above, we examined the hypothesis that the distribution of connectivity degrees for Kenyon cells would match the flexibility of those cells under homeostatic or resource constraints on their total synaptic weight. We used a simple measure of flexibility: the size of the allowed synaptic weight configuration space (Fig 1b and 1c). We next considered a related but more restricted hypothesis: that KCs directly maximize their flexibility. For each constraint, we maximized the size of the allowed synaptic weight space to find the optimal degrees.

For the bounded net weight constraint (Eq 1), this consists of maximizing the volume under the simplex (Eq 2) and is equivalent to finding the mode of the zero-truncated Poisson distribution. We found an approximately linear relationship between the optimal degree and the maximum net connection strength:

$$K_V^* = \alpha\bar{S} - \frac{1}{2} + \mathcal{O}\left(1/K_V^*\right) \tag{14}$$

The derivation of this equation involves the harmonic numbers, which are defined by positive integers, so it applies only for $K_V^* \geq 1$ (Optimal degrees: Bounded net weight). Under the fixed net weight constraint, we similarly found an approximately linear relationship between the optimal degree and the net connection strength (Optimal degrees: fixed net weight):

$$K_A^* = \alpha\bar{S} + 2 + \mathcal{O}\left(1/K_A^*\right) \tag{15}$$

This equation applies only for $K \geq 2$, for a similar reason as above (Optimal degrees: fixed net weight). By comparing $K_V^*$ and $K_A^*$, we see that the optimally flexible degrees under the fixed net weight constraint are higher than those for the bounded net weight constraint. Eqs (14) and (15) reveal that to leading order, the model comparisons of Figs 3–5 encapsulate linear fits of $K$ as a function of $\bar{S}$ while accounting for the variability around that line predicted by each constraint.

In both larval and adult KCs, we observed approximately linear relationships between the total number of synapses and number of partners for each KC type (Fig 6). To quantify this linear relationship, we computed the Pearson correlation between the number of synapses and number of partners for the different KC types. In the larva, we found that less mature KCs better matched this linear relationship (Table 1: young > multi-claw > single-claw). The same was true for the adult KCs (Table 2: alpha prime > core > surface > posterior).

The binomial model we examined above does not depend on or model synapse counts. If it were augmented with a wiring process where each connection independently sampled a number of synapses, its total synapse count and number of connections would also be linearly

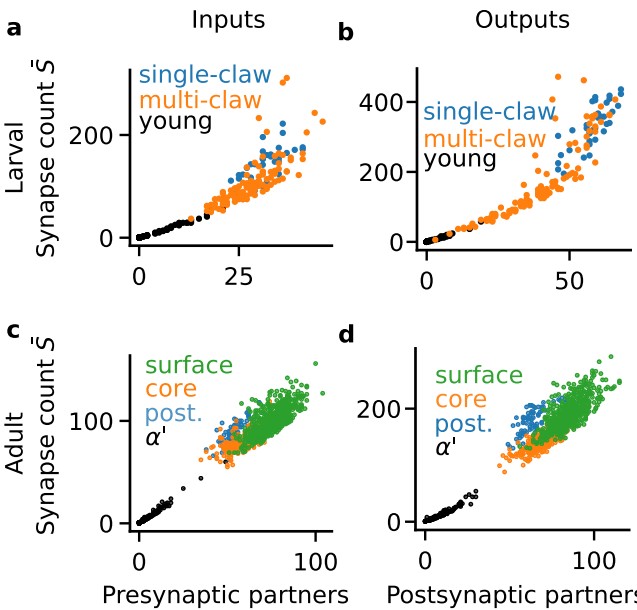

**Fig 6. Relation between number of synaptic partners and synapse counts in *D. melanogaster* Kenyon cells. (a)** Inputs to Kenyon cells (KCs) of the first instar larva. (**b**) Outputs of the first instar KCs. (**c**) Inputs to adult KCs in the $\alpha$ lobe. (**d**) Outputs of adult KCs in the $\alpha$ lobe.

related. Each of these three models are thus consistent with the same qualitative result. The model comparison we performed above tests which best explains the data, accounting for the variability around the mode described by each model (Figs 3–5).

Finally, we generalized this optimization to allow the constraints on total synaptic weights to explicitly depend on the number of inputs:

$$\sum_{i=1}^{K} J_i \leq \bar{J} K^p \text{ or } \sum_{i=1}^{K} J_i = \bar{J} K^p \tag{16}$$

for the bounded and fixed net weight constraints, respectively. This scaling of the summed synaptic weight corresponds to scaling the individual synaptic weights as $K^{p-1}$. If every synaptic weight has an order $1/K$ strength, the sum of the synaptic weights would be order 1 and $p = 0$. If every synaptic weight has an order 1 strength, the summed weight is order $K$ and $p = 1$. If synaptic weights have balanced $\left(1/\sqrt{K}\right)$ scaling [48], then the summed weight would have $p = 1/2$. Under this generalization of our constraint models, our Bayesian model comparisons still apply if we take the total synaptic weight to be proportional to the number of synapses: $K^p \bar{J} = \alpha \bar{S}$ instead of Eq 10. That corresponds to the requirement that the scaling of synaptic weights with the number of inputs does not arise from scaling the number of synapses, but from other physiological mechanisms. This generalization still led to approximately linear relationships between the optimal degree and the total synaptic weight (Optimal

**Table 1. Correlation of number of synapses and number partners in larval Kenyon cells.**

| Cell type | Inputs | Outputs |
|---|---|---|
| Young | 0.99 | 0.99 |
| Multi-claw | 0.77 | 0.86 |
| Single-claw | 0.67 | 0.8 |

**Table 2. Correlation of number of synapses and number partners in adult alpha lobe Kenyon cells.**

| Cell type | Inputs | Outputs |
|---|---|---|
| alpha prime | 0.99 | 0.97 |
| core | 0.76 | 0.81 |
| surface | 0.74 | 0.74 |
| posterior | 0.69 | 0.69 |

degrees: Bounded net weight and Optimal degrees: fixed net weight):

$$
\begin{aligned}
K_V^* &= \alpha \exp(p)\bar{S} - \frac{1}{2} + \mathcal{O}(1/K_V^*) \\
K_A^* &= \alpha \exp(p)\bar{S} + 2 - p + \mathcal{O}(1/K_A^*)
\end{aligned}
\tag{17}
$$

As before, we see that the optimally flexible degree under the fixed net weight constraint, $K_A^*$, is greater than that under the bounded net weight constraint, $K_V^*$. In this generalization, we can make a similar assumption as before to relate the net synaptic weight $\bar{J}$ to anatomical measures of connection strength. If we assume that $K^p \bar{J} = \alpha \bar{S}$ so that the number of synapses $\bar{S}$ absorbs the scaling with $K^p$, consistent with its origin reflecting the size of a neuron, the same analysis and results of Figs 3–5 follow. If we instead assumed that $\bar{J} = \alpha \bar{S}$, so that the anatomically measured total synaptic weight were $K^p \alpha \bar{S}$, a model comparison that also accounts for the unknown parameter $p$ would be required.

## Discussion

We hypothesized that under a particular constraint, the probability of a neuron having $K$ synaptic partners is proportional to the size of the space of allowed circuit configurations with $K$ partners. The general idea of considering the space of allowed configurations can be traced back to Elizabeth Gardner's pioneering work examining the associative memory capacity of a perceptron for random input patterns [49]. In the limit of infinitely many connections and input patterns, that model yields predictions for the distributions of synaptic weights [8–11]. Here, in contrast, we examined the hypothesis that the size of the space of allowed configurations—the flexibility of a neuron's connectivity under constraint—governs the distribution of the number of connections without defining a computational task. This motivated predictions for neural degree distributions, rather than synaptic weight distributions. We examined constraints on the total strength of connections to or from a neuron and found that overall, the degree distribution corresponding to flexible connectivity under a homeostatically fixed total connection strength gave the best explanation for mushroom body connectivity.

### Flexible connectivity and circuit development

Computational flexibility should be desirable for an organism's fitness, allowing the organism to solve problems in a variety of environments. One mechanism of adaptability and flexibility is to build the nervous system out of computationally flexible units that may over time adapt to specific computational roles. Our results are suggestive that this type of strategy may be at play in the development of mushroom body connectivity in the first instar *D. melanogaster* larva. The log odds for larval young KCs vastly favor the constraints models over the binomial model (Figs 3c and 4c). The odds also approximately decreasingly favor

the constraint model with KC maturity (Figs 3 and 4). In the adult, the log odds favored flexible connectivity under constraints on the net synaptic weight over the binomial random wiring model most for the alpha prime KCs, and more for core than surface KCs (Figs 3f and 4f, Figures D, E in S1 Figs). These suggest that flexibility under constraints might also reflect a developmental or experience-dependent progression in the alpha lobe KCs, but it remains a better explanation for their connectivity than binomial wiring even in the more mature KCs of the adult. The less mature KCs in the larva and adult also showed more linear relationships between their number of synapses and number of synaptic partners (Table 1), better matching the prediction of maximizing the space of allowed configurations under a constraint. Together, these results suggest that Kenyon cell connectivity is structured to be flexible early in development, allowing many possible connectivity configurations to support specialization as the organism matures.

## Anatomical measures of connection strength

To test the hypothesis that neurons in the mushroom body are subject to a pressure towards flexible connectivity under constraints, we required measurements of the total input or output connection strength of these neurons. For this purpose, we used electron microscopic reconstructions of mushroom body circuitry [38, 39]. These published data contain anatomical measurements of connectivity: the number of synapses between neurons. The general types of constraint we considered (bounded or homeostatically fixed total connection strengths) might not operate directly on synapse counts. To account for this uncertainty, we assumed that synapse counts were proportional to the constrained total connection strength (Eq 10) [38]. Spatially detailed, biophysical neuron models could in principle be used to account for synapse locations and the passive and active membrane conductances transforming anatomical connectivity into physiological connection strengths in specific neurons. In hippocampal pyramidal cells, cerebellar Purkinje cells, and Drosophila visual neurons, dendritic structures can compensate for signal decay systems [50–53]. If this is also the case in mushroom body neurons, detailed spatial models to relate anatomical and physiological connection strengths might not provide additional insight. Alternatively, additional information about the processes governing homeostatic synaptic scaling or synaptic resource limits could motivate models of a different functional form than Eq 10 [54, 55].

## Physiological constraints on neural circuits

We modeled constraints as requirements on synaptic weight vectors, consistent with point neuron models commonly used in studies of neural computation, rather than specifying the biophysical implementation of these constraints. Minimizing the amount of wire used to connect neural circuits can predict the spatial layout of neural systems (e.g., [56–60]) and dendritic arborizations [61, 62]. We examined setting the number of connections separately from the strengths of connections, consistent with the assumption that rewiring neural circuits is more costly than changing the strength of existing connections [63].

Neural activity faces metabolic constraints [64]. In early sensory systems, the combination of metabolic constraints with sensory encoding needs can explain the structure of neural activity [65–69]. In our model, both wiring and metabolic costs could be related to setting the parameter $\bar{J}$. We hope that, analogously to how metabolic costs and encoding performance combine in metabolically efficient coding, the idea of flexibility under constraints might be useful in determining how metabolic and wiring constraints interact with computational tasks to shape neural circuit structures.

## Materials and methods

### Measuring synaptic weight configuration spaces

First, consider measuring the available configurations of synaptic weights when they can vary continuously. Consider the total synaptic weight, divided into $K$ segments (Fig 7). For one synaptic weight, the measure of the weight configurations is

$$\int_0^{\bar{J}} dJ_1 = \bar{J} \tag{18}$$

For two synaptic weights, the available configurations are measured by

$$\int_0^{\bar{J}} \int_{J_1}^{\bar{J}} dJ_2 \, dJ_1 = \frac{\bar{J}^2}{2} \tag{19}$$

and in general,

$$\int_0^{\bar{J}} \int_{J_1}^{\bar{J}} \int_{J_2}^{\bar{J}} \cdots \int_{J_{K-1}}^{\bar{J}} dJ_K \cdots dJ_3 \, dJ_2 \, dJ_1 = \frac{\bar{J}^K}{K!} \tag{20}$$

which is the volume under the simplex with vertex length $\bar{J}$.

Now consider synaptic weights that vary discretely by $\Delta J$, with $\bar{J} = M\Delta J$. How many ways can we assign $M$ units of synaptic weight amongst $K$ partners? For $K = 1$, this is $M$. For $K = 2$, this is

$$\sum_{n_1=1}^{M} \sum_{n_2=n_1+1}^{M} = \frac{M(M-1)}{2} \tag{21}$$

and in general the number of combinations of $M$ units of synaptic weight in $K$ connections is given by the binomial coefficient

$$\binom{M}{K} = \frac{M!}{K!(M-K)!} \tag{22}$$

**a**

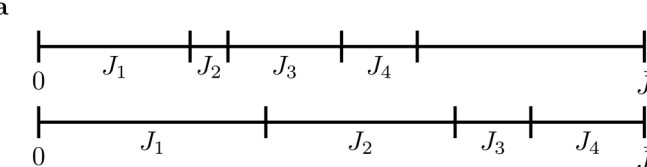

**b**

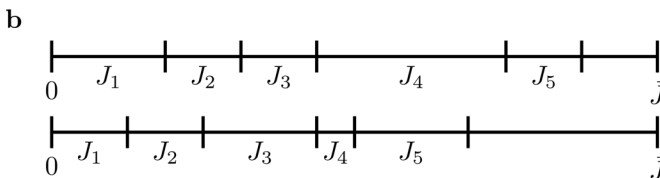

**Fig 7. Synaptic weight configurations. a)** Two example configurations of $K = 4$ synaptic weights with $J_1 + J_2 + J_3 + J_4 \le \bar{J}$. **b)** Two examples of $K = 5$ synaptic weights with sum bounded by $\bar{J}$.

For large $M$, the binomial coefficient is

$$\frac{M^K}{K!}(1 + \mathrm{O}(1/M)) \tag{23}$$

Now if we measure synaptic weights relative to $\Delta J$, this replaces $\bar{J}$ with $\bar{J}/\Delta J = M$ in the simplex's volume. So for large $M$, the volume under the simplex approximates the number of allowed configurations.

Similarly, the surface area of the simplex (Eq 7) approximates the number of allowed configurations under the fixed net weight constraint (Eq 6) if we discretize the synaptic weights. It is measured by the $K - 1$ dimensional Haussdorff measure (hyper-surface area). We can compute the surface area by differentiating the volume (Eq 2) with respect to the inner radius of the simplex (the minimal distance from the origin to its surface) [70]. For the regular simplex with vertices at $\bar{J}$, that inner radius is $r = \bar{J}/\sqrt{K}$. Differentiating the volume with respect to $r$ thus yields the surface area

$$
\begin{aligned}
A \quad &= \frac{dV}{dr} \\
&= \sqrt{K}\frac{dV}{d\bar{J}} \\
&= \frac{\bar{J}^{K-1}\sqrt{K}}{(K-1)!}
\end{aligned}
\tag{24}
$$

Note, however, that the inner radius and thus the surface area depends on the norm of the space of synaptic weight configurations (Distances in synaptic configuration space).

## Model comparison

Under equal prior likelihoods for two models $X$ and $Y$, the posterior likelihood ratio between two models, $X$ and $Y$ is

$$\frac{p_X(K|\bar{S})}{p_Y(K|\bar{S})} = \frac{\int d\alpha \prod_i p_X(K_i|\bar{S}_i, \alpha)p(\alpha)}{\int d\alpha \prod_i p_Y(K_i|\bar{S}_i, \alpha)p(\alpha)} \tag{25}$$

where $i$ indexes data points. We consider the Laplace approximations for the posterior odds, obtained by writing $p_X = \exp \ln p_X$ and Taylor expanding the log likelihood $\ln p_X$ in $\alpha$ around its maximum likelihood value,

$$\hat{\alpha} = \arg \max_{\alpha} \prod_i p_X(K_i|\bar{S}_i, \alpha) \tag{26}$$

Truncating at second order then yields a tractable Gaussian integral over the unknown parameter:

$$
\begin{aligned}
\int d\alpha \, p(\alpha) \prod_i p_X(K_i|\bar{S}_i, \alpha) \quad &= \int d\alpha \, p(\alpha) \prod_i \exp \ln p_X(K_i|\bar{S}_i, \alpha) \\
&\approx \int d\alpha \, p(\alpha) \prod_i p_X(K_i|\bar{S}_i, \hat{\alpha}) \exp \frac{-(\alpha - \hat{\alpha})^2}{2\sigma_i^2}
\end{aligned}
\tag{27}
$$

where the integrals run over the allowed range for $\alpha$ and

$$\sigma_i^2 = -\frac{\partial^2}{\partial\alpha^2}\ln\,p_X(K_i|\bar{S}_i,\alpha)\Big|_{\hat{\alpha}} \tag{28}$$

Under a flat prior for non-negative $\alpha$, the marginal likelihood is:

$$p_X(K|\bar{S}) \approx \left(\prod_i p_X(K_i|\bar{S}_i,\hat{\alpha})\right)\sqrt{\frac{\pi\sigma^2}{2}}\left(1 + \text{Erf}\left(\frac{\hat{\alpha}}{\sqrt{2\sigma^2}}\right)\right) \tag{29}$$

where $1/\sigma^2 = \sum_i 1/\sigma_i^2$. The simplex volume distribution is a truncated Poisson; we might reasonably use the Jeffreys prior for the Poisson distribution, $p(\alpha) \propto 1/\sqrt{\alpha}$. In that case, the marginal likelihood is

$$p_X(K|\bar{S}) \approx \left(\prod_i p_X(K_i|\bar{S}_i,\hat{\alpha})\right)\frac{\pi\sqrt{\hat{\alpha}}}{2}\exp\left(-\frac{\hat{\alpha}^2}{4\sigma^2}\right)\left(\mathcal{I}_{-\frac{1}{4}}\left(\frac{\hat{\alpha}^2}{4\sigma^2}\right) + \mathcal{I}_{\frac{1}{4}}\left(\frac{\hat{\alpha}^2}{4\sigma^2}\right)\right) \tag{30}$$

where $\mathcal{I}_a$ is the modified Bessel function of the first kind. We will drop the indices on $K, \bar{S}$ in most of the remaining sections, reintroducing them where necessary.

## Model comparison: Bounded net weight model

Under a bounded net weight, the degree distribution is:

$$p_V(K|\bar{S},\alpha) = \frac{(\alpha\bar{S})^K}{Z_V(\bar{S},\alpha)K!} \tag{31}$$

The normalization constant $Z$ is

$$Z_V(\bar{S},\alpha) = \sum_{K=1}^{\infty}\frac{(\alpha\bar{S})^K}{K!} = \exp(\alpha\bar{S}) - 1 \tag{32}$$

so the simplex volume distribution is a zero-truncated Poisson distribution. We will make a Laplace approximation for the simplex volume distribution around $\hat{\alpha}$, leading to the posterior odds Eq 29 (for a flat prior on non-negative $\alpha$) or Eq 30 (for the Poisson Jeffreys prior). To calculate the Laplace approximation for the posterior odds we need $\hat{\alpha}$ and $\sigma^2$. The derivatives of $\ln p_V$ can be calculated directly (again dropping indices over measurements),

$$\begin{aligned}\frac{\partial}{\partial\alpha}\ln p_V(K|\bar{S},\alpha) &= \frac{K}{\alpha} + \bar{S}\left(\frac{1}{1-\exp\alpha\bar{S}} - 1\right)\\\frac{\partial^2}{\partial\alpha^2}\ln p_V(K|\bar{S},\alpha) &= -\frac{K}{\alpha^2} + \left(\frac{\bar{S}}{2}\text{Csch}\left(\frac{\alpha\bar{S}}{2}\right)\right)^2\end{aligned} \tag{33}$$

So we have

$$\sigma^2 = \left(\sum_i\left(\frac{K_i}{\hat{\alpha}^2} - \left(\frac{\bar{S}_i}{2}\text{Csch}\left(\frac{\hat{\alpha}\bar{S}_i}{2}\right)\right)^2\right)^{-1}\right)^{-1} \tag{34}$$

and the maximum likelihood solution for $\alpha$ satisfies

$$0 = \sum_i \frac{K_i}{\hat{\alpha}} - \frac{\bar{S}_i}{1 - \exp \hat{\alpha} \bar{S}_i} \tag{35}$$

## Model comparison: Fixed net weight model

Under the fixed net synaptic weight, our model is that the degree distribution is proportional to the surface area of the simplex:

$$p_A(K|\bar{S}, \alpha) = \frac{1}{Z_A(\bar{S}, \alpha)} \frac{(\alpha\bar{S})^{K-1}\sqrt{K}}{(K-1)!} \tag{36}$$

where

$$Z_A(\bar{S}, \alpha) = \sum_{K=1}^{\infty} \frac{(\alpha\bar{S})^{K-1}\sqrt{K}}{(K-1)!} \tag{37}$$

To calculate $\hat{\alpha}$ and $\sigma^2$ we need the derivatives of $\ln p_A$.

$$\frac{\partial}{\partial\alpha} \ln p_A = \frac{\partial}{\partial\alpha} \ln A - \frac{\partial}{\partial\alpha} \ln Z \tag{38}$$

where

$$\frac{\partial}{\partial\alpha} \ln A = \frac{K-1}{\alpha} \tag{39}$$

and we use the identity

$$\frac{\partial}{\partial\alpha} \ln Z = \frac{\frac{\partial}{\partial\alpha} Z}{Z} \tag{40}$$

We next bound $\frac{\partial}{\partial\alpha} Z$.

$$\begin{aligned} \frac{\partial}{\partial\alpha} Z &= \sum_{K=2}^{\infty} \frac{(\alpha\bar{S})^{K-2}}{(K-2)!} \sqrt{K} S \\ &= S\sum_{K=1}^{\infty} \frac{(\alpha\bar{S})^{K-1}}{(K-1)!} \sqrt{K}\sqrt{1 + \frac{1}{K}} \end{aligned} \tag{41}$$

For $K \geq 1$, $\sqrt{1 + 1/K}$ is bounded above by $\sqrt{2}$ and below by 1. So,

$$SZ < \frac{\partial}{\partial\alpha} Z < \sqrt{2}SZ \tag{42}$$

Inserting these into the critical point equation for $\hat{\alpha}$ provides the bounds:

$$\frac{\sum_i(K_i - 1)}{\sqrt{2}\sum_i \bar{S}_i} < \hat{\alpha} < \frac{\sum_i(K_i - 1)}{\sum_i \bar{S}_i} \tag{43}$$

We will also need the curvature of $\ln p_A$ w.r.t. $\alpha$ at $\hat{\alpha}$:

$$\sigma^2 = -\frac{\partial^2}{\partial\alpha^2} \ln p_A \tag{44}$$

Similarly to the first derivative,

$$\frac{\partial^2}{\partial \alpha^2} \ln p_A = \frac{\partial^2}{\partial \alpha^2} \ln A - \frac{\partial^2}{\partial \alpha^2} \ln Z \tag{45}$$

where

$$\frac{\partial^2}{\partial \alpha^2} \ln A(K|\bar{S}, \alpha) = -\frac{(K-1)}{\alpha^2} \tag{46}$$

We use the identity

$$\frac{\partial^2}{\partial \alpha^2} \ln Z = \frac{\frac{\partial^2}{\partial \alpha^2} Z}{Z} - \frac{\left(\frac{\partial}{\partial \alpha} Z\right)^2}{Z^2} \tag{47}$$

The curvature of $Z$ is

$$\begin{aligned}
\frac{\partial^2}{\partial \alpha^2} Z \quad &= S^2 \sum_{K=3}^{\infty} \frac{(\alpha \bar{S})^{K-3}}{(K-3)!} \sqrt{K} \\
&= S^2 \sum_{K=1}^{\infty} \frac{(\alpha \bar{S})^{K-1}}{(K-1)!} \sqrt{K} \sqrt{1 + \frac{2}{K}}
\end{aligned} \tag{48}$$

The final term $\sqrt{1 + \frac{2}{K}}$ is bounded above by $\sqrt{3}$ and below by 1, so

$$S^2 Z < \frac{\partial^2}{\partial \alpha^2} Z < \sqrt{3} S^2 Z \tag{49}$$

Defining upper and lower bounds for $\frac{\partial^2}{\partial \alpha^2} \ln Z$ using the upper and lower bounds of the first and second terms in Eq 47 yields:

$$-S^2 < \frac{\partial^2}{\partial \alpha^2} \ln Z < (\sqrt{3} - 1)S^2 \tag{50}$$

The upper bound for $\frac{\partial^2}{\partial \alpha^2} \ln Z$ provides an upper bound for $\sigma^2$, while neglecting $Z$ provides a lower bound for $\sigma^2$ (since $Z \geq 1$ from Eq 37, so that $\ln Z \geq 0$):

$$\frac{K-1}{\alpha^2} \leq \sigma^2 < \frac{K-1}{\alpha^2} + (\sqrt{3} - 1)S^2 \tag{51}$$

The posterior odds for the simplex area are:

$$\int d\alpha \, p(\alpha) \prod_i p_A(K_i|\bar{S}_i, \alpha) \approx \left( \prod_i p_A(K_i|\bar{S}_i, \hat{\alpha}) \right) \int d\alpha \, p(\alpha) \exp\left( \frac{(\alpha - \hat{\alpha})^2}{2\sigma^2} \right) \tag{52}$$

where $\sigma^2 = 1/\sum_i 1/\sigma_i^2$. We use the upper and lower bounds for $\sigma_i^2$ to define upper and lower bounds, respectively, for the likelihood's variance:

$$\begin{aligned}
\sigma_U^2 \quad &= \left( \sum_i \left( \frac{(K_i - 1)}{\hat{\alpha}^2} + \left(\sqrt{3} - 1\right)\bar{S}_i^2 \right)^{-1} \right)^{-1} \\
\sigma_L^2 \quad &= \left( \sum_i \left( \frac{(K_i - 1)}{\hat{\alpha}^2} \right)^{-1} \right)^{-1}
\end{aligned} \tag{53}$$

We compute $\hat{\alpha}$ numerically by maximizing the likelihood, and compute $p_A(K_i|\bar{S}_i, \hat{\alpha})$ also numerically, estimating $Z$ by ranging over $K = 1$ to $2 \max_i \bar{S}_i$.

## Bounds for the posterior odds of the fixed net weight model

The derivative of the posterior odds under the flat prior, Eq 29, with respect to $\sigma$ is proportional to

$$1 - \sqrt{\frac{2}{\pi}}\frac{\alpha}{\sigma}\exp\left(-\frac{\alpha^2}{2\sigma^2}\right) + \text{Erf}\left(\frac{\alpha}{\sqrt{2}\sigma}\right) \tag{54}$$

Since $\alpha > 0$ and $\sigma > 0$, the last term is bounded between 0 and 1. The middle term is proportional to the form $x \exp(-x^2/2)$, which is maximized by $1/\sqrt{e}$ at $x = 1$. Since $\sqrt{2/\pi e} < 1$, the middle term is less than 1 and the derivative of the posterior odds under a flat prior for $\alpha$, with respect to $\sigma$, is non-negative. The upper bound for $\sigma^2$ thus provides an upper bound on the posterior odds. We see that the posterior likelihood $\prod_i p_A(K_i|\bar{S}_i)$ increases from $\sigma_L^2$ to $\sigma_U^2$ (reflected in the log posterior odds ratio for the simplex volume vs the simplex area, Figures A, B, D-I in S1 Figs).

The derivative of the posterior odds under the Poisson Jeffreys prior, Eq 30, with respect to $\sigma^2$, is proportional to

$$-\frac{\alpha^2}{2\sigma^3}\exp\left(-\frac{\alpha^2}{4\sigma^2}\right)\left( \mathcal{I}_{-\frac{5}{4}}\left(\frac{\alpha^2}{4\sigma^2}\right) + \mathcal{I}_{-\frac{3}{4}}\left(\frac{\alpha^2}{4\sigma^2}\right) + 2\mathcal{I}_{-\frac{1}{4}}\left(\frac{\alpha^2}{4\sigma^2}\right) \right.$$
$$\left. +2\mathcal{I}_{\frac{1}{4}}\left(\frac{\alpha^2}{4\sigma^2}\right) + \mathcal{I}_{\frac{3}{4}}\left(\frac{\alpha^2}{4\sigma^2}\right) + \mathcal{I}_{\frac{5}{4}}\left(\frac{\alpha^2}{4\sigma^2}\right) \right) \tag{55}$$

We saw that the posterior odds for the simplex area distribution also increased with $\sigma^2$ for the Poisson Jeffreys prior (S1 Figs).

## Model comparison: Zero-truncated binomial model

The marginal likelihood for the zero-truncated binomial with distribution $p_B$ is

$$\int dq\, p(q)\prod_i p_B(K_i|N, q) \approx \left(\prod_i p_B(K_i|N, \hat{q})\right)\int dq\, p(q)\prod_i \exp\left(-\frac{(q-\hat{q})^2}{2\sigma^2}\right) \tag{56}$$

where $p_B(K_i|N, q)$ is given by Eq 11. For connections to larval KCs, we used the total number of traced projection neurons (PNs) and KCs as the binomial parameter $N$, averaged over the two sides of the brain [38]. For projections from larval KCs, we used the total number of KCs and output neurons, averaged over the two sides, as $N$. For projection to adult KCs, we used the number of Kenyon cells plus 150 (the estimated number of olfactory PNs) as $N$ [71]. For projections from adult KCs, we used the number of KCs and output neurons labelled in the data as $N$.

The variance with respect to $q$ is determined as in Eq (28). The derivates of $\ln p_B$ are, again dropping indices on $K$,

$$
\begin{aligned}
\frac{\partial}{\partial q} \ln p_B &= \frac{K}{q} - \frac{N-K}{1-q} - \frac{N(1-q)^{N-1}}{1-(1-q)^N} \\
\frac{\partial^2}{\partial q^2} \ln p_B &= -\frac{K}{q^2} + \frac{N-K}{(1-q)^2} + \frac{N(N-1+(1-q)^N)(1-q)^{N-2}}{((1-q)^N-1)^2}
\end{aligned}
\tag{57}
$$

The maximum likelihood parameter $\hat{q}$ for the zero-truncated binomial, with $M$ samples of $K$, each with $N$ trials, obeys:

$$
\frac{\hat{q}}{1-(1-\hat{q})^N} = \frac{\sum_{i=1}^{M} K_i}{MN}
\tag{58}
$$

and the variance at $\hat{q}$ is

$$
\sigma^2 = \left( \sum_i \left( \frac{K_i}{\hat{q}^2} - \frac{N-K_i}{(1-\hat{q})^2} - \frac{N(N-1+(1-\hat{q})^N)(1-\hat{q})^{N-2}}{((1-\hat{q})^N-1)^2} \right)^{-1} \right)^{-1}
\tag{59}
$$

## Optimal degrees: Bounded net weight

Now we examine what numbers of synaptic partners maximize the size of the allowed configuration space under the bounded net weight constraint. Here we generalize the constraint to allow the maximum total synaptic weight to explicitly depend on the number of inputs, $K$:

$$
\sum_{i=1}^{K} J_i \leq \bar{J} K^p
\tag{60}
$$

where we will typically take $0 \leq p \leq 1$. This replaces the maximum weight $\bar{J}$ with $K^p \bar{J}$ in the volume:

$$
V = \frac{(\bar{J} K^p)^K}{K!}
\tag{61}
$$

The volume is non-decreasing in $\bar{J}$. We compute its derivative with respect to $K$ by analytically continuing the factorial to real values of $K$ as the Gamma function, yielding

$$
\begin{aligned}
\frac{\partial V}{\partial K} &= \frac{(\bar{J} K^p)^K}{K!} \left( \ln \bar{J} K^p + p + \gamma - H_K \right) \\
&= \frac{(\bar{J} K^p)^K}{K!} \left( \ln \bar{J} K^p + p - \ln K - \frac{1}{2K} + \mathcal{O}(K^{-2}) \right)
\end{aligned}
\tag{62}
$$

where $H_K$ is the $K$th harmonic number,

$$
H_K = \sum_{x=1}^{K} \frac{1}{x}
\tag{63}
$$

We used Euler's expansion for the harmonic numbers,

$$H_K = \gamma + \ln K + \frac{1}{2K} + \mathcal{O}(K^{-2}) \tag{64}$$

At a critical point in $K$, truncating $\mathcal{O}(1/K^2)$ and higher-order terms, we find

$$
\begin{aligned}
\bar{J}(K^*)^p & = e^{\frac{1}{2K^*} - p}(K^*) + \mathcal{O}(1/K^2) \\
& = \left(K^* + \frac{1}{2}\right) \exp\left(-p\right) + \mathcal{O}(1/K^*)
\end{aligned}
\tag{65}
$$

Substituting $K^p \bar{J} = \alpha \bar{S}$ yields

$$K^* = \alpha \exp\left(p\right)\bar{S} - \frac{1}{2} + \mathcal{O}(1/K^*) \tag{66}$$

Alternatively, the critical point can be calculated without first extending $K$ to real numbers by using ratios:

$$1 = \frac{V(K^*)}{V(K^* + 1)} = \frac{(K^* + 1)^{1-p}}{\bar{J}} \left(\frac{K^*}{K^* + 1}\right)^{pK^*} \tag{67}$$

which yields

$$
\begin{aligned}
\bar{J}(K^*)^p & = (K^* + 1)\left(\frac{K^*}{K^* + 1}\right)^{p(K^* + 1)} \\
& = (K^* + 1 - p) \exp\left(-p\right) + \mathcal{O}(1/K^*)
\end{aligned}
\tag{68}
$$

## Optimal degrees: Fixed net weight

If $\sum_j J_j = \bar{J}K^p$ then we have the surface area of the $K - 1$ simplex. We consider a regular simplex (equal side lengths) with vertex length $\bar{J}K^p$ (from the origin to any vertex). Its surface area is

$$A = \frac{(\bar{J}K^p)^{K-1}\sqrt{K}}{(K-1)!} \tag{69}$$

By the same method as above, the derivative with respect to $K$ is

$$
\begin{aligned}
\frac{\partial A}{\partial K} & = \frac{(\bar{J}K^p)^{K-1}}{2\sqrt{K}(K-1)!} \left(2K \ln\left(\bar{J}K^p\right) + 2(K-1)p - 2K(H_{K-1} - \gamma) + 1\right) \\
& = \frac{(\bar{J}K^p)^{K-1}}{2\sqrt{K}(K-1)!} \Bigg( 2K \ln\left(\bar{J}K^p\right) + 2(K-1)p \\
& \quad - 2K\left(\ln\left(K-1\right) + \frac{1}{2(K-1)} + \mathcal{O}\left(\frac{1}{K^2}\right)\right) + 1\Bigg)
\end{aligned}
\tag{70}
$$

Since $H_{K-1}$ appears in the derivative, we only consider the derivative at $K \geq 2$. At a critical point in $K$, dividing through by $K$ and truncating $\mathcal{O}(1/K^2)$ and higher-order terms yields

$$
\begin{aligned}
\ln \bar{J}K^p \quad &= -p\frac{K-1}{K} + \ln(K-1) - \frac{1}{2K} - \frac{1}{2(K-1)} + \mathcal{O}(1/K^2) \\
\bar{J}K^p \quad &= (K-1)\exp(-p)\exp\left(p/K - \frac{1}{2K} - \frac{1}{2(K-1)} + \mathcal{O}(1/K^2)\right) \\
&= \exp(-p)(K+p-2) + \mathcal{O}(1/K)
\end{aligned}
\tag{71}
$$

and substituting $K^p\bar{J} = \alpha\bar{S}$ yields

$$
K^* = \alpha\exp(p)\bar{S} + 2 - p + \mathcal{O}(1/K^*)
\tag{72}
$$

If we do not continue to real $K$ first, we instead have

$$
1 = \frac{A(K^*)}{A(K^*+1)} = \frac{(K^*)^{1-p}}{\bar{J}}\left(\frac{K^*}{K^*+1}\right)^{pK^*+\frac{1}{2}}
\tag{73}
$$

which yields

$$
\begin{aligned}
\bar{J}(K^*)^p \quad &= K^*\left(\frac{K^*}{K^*+1}\right)^{pK^*+\frac{1}{2}} \\
&= \left(K^* + \frac{p-1}{2}\right)\exp(-p) + \mathcal{O}(1/K^*)
\end{aligned}
\tag{74}
$$

## Distances in synaptic configuration space

Above we assumed that synaptic weight configurations could travel between different points in the synaptic weight space along straight lines, endowing the $K$-dimensional synaptic weight space with a Euclidean (or 2-) norm. This amounts to assuming that synaptic weights can vary together. This could be interpreted, for example, as allowing a unit of synaptic weight (a receptor, perhaps) to be transferred directly between connections. An alternative is to assume that synaptic weights must move separately, which corresponds endowing the synaptic weight space with the 1-norm. In the above interpretation this would mean separating the removal of a receptor from one synapse from the addition of a receptor to another synapse. This changes the surface area of the simplex, since its inner radius is $\bar{J}/K$ rather than $\bar{J}/\sqrt{K}$:

$$
A_1 \quad = \frac{(\bar{J}K^p)^{K-1}K}{(K-1)!}
\tag{75}
$$

Changing the norm for the synaptic weights leaves the above calculation of the posterior odds for the fixed net weight model mostly unchanged. The factors of $\sqrt{K}$ in the normalization constant are replaced by $K$; this removes the square roots in the derivation of the upper bound for the variance with respect to $\alpha$ so that

$$
\frac{K-1}{\alpha^2} \leq \sigma^2 \quad < \frac{K-1}{\alpha^2} + 2S^2
\tag{76}
$$

The optimal number of connections can be calculated in the same manner as previously. The derivative of $A_1$ with respect to $K$ is (to order $1/K$):

$$\frac{\partial A_1}{\partial K} \approx \frac{(\bar{J}K^p)^{K-1}}{(K-1)!}\left( K\ln\bar{J}K^p + (K-1)p - K\left(\ln(K-1) + \frac{1}{2(K-1)} + \mathcal{O}\left(\frac{1}{K^2}\right)\right) + 1\right) \quad (77)$$

At a critical point in $K$, truncating $\mathcal{O}(1/K^2)$ and higher-order terms yields

$$\bar{J} \approx K^{-p}(K-1)\exp\left(\frac{1}{K(K-1)} - p\frac{K-1}{K}\right) \quad (78)$$

## Supporting information

**S1 Figs. Supporting figures.**
(PDF)

## Acknowledgments

We thank Ramakrishnan Iyer, Casey Schneider-Mizell, and Saskia de Vries for helpful discussions. We wish to thank the Allen Institute founder, Paul G. Allen, for his vision, encouragement and support.

## Author Contributions

**Conceptualization:** Gabriel Koch Ocker, Michael A. Buice.

**Formal analysis:** Gabriel Koch Ocker.

**Investigation:** Gabriel Koch Ocker.

**Software:** Gabriel Koch Ocker.

**Supervision:** Michael A. Buice.

**Visualization:** Gabriel Koch Ocker.

**Writing – original draft:** Gabriel Koch Ocker.

**Writing – review & editing:** Michael A. Buice.

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
