## [Decision Letter · Decision Letter 0]

29 Oct 2019

Dear Dr Ocker,

Thank you very much for submitting your manuscript 'Flexible neural connectivity under constraints on total connection strength' for review by PLOS Computational Biology. Your manuscript has been fully evaluated by the PLOS Computational Biology editorial team and in this case also by independent peer reviewers. The reviewers appreciated the attention to an important problem, but raised some substantial concerns about the manuscript as it currently stands. While your manuscript cannot be accepted in its present form, we are willing to consider a revised version in which the issues raised by the reviewers have been adequately addressed. We cannot, of course, promise publication at that time.

Sincerely,

Jeff Beck

Associate Editor

PLOS Computational Biology

Lyle Graham

Deputy Editor

PLOS Computational Biology

[LINK]

Reviewer's Responses to Questions

**Comments to the Authors:**

Reviewer #1: Neuronal computation operates under certain biological constraints. How brain centers are organized to deal with these constraints while maximizing their performance is poorly known. In this manuscript, Ocker and Buice use the Drosophila mushroom body to address this question. The mushroom body is an associative center that has been primarily studied for its role during olfactory memory formation. The mushroom body receives direct input from the olfactory pathway: The olfactory sensory neurons (the first-order neurons) each expresses only one olfactory receptor gene and the neurons expressing the same receptor converge in the antennal lobe where they innervate the same glomerulus. The projection neurons (the second-order neurons) connect individual glomeruli to different third-order neurons, namely the Kenyon cells of the mushroom body. Because a given odorant molecule binds to different olfactory receptors, most odors activate a combination of antennal lobe glomeruli. These broad activity patterns are transformed by the mushroom body into sparser ones: each odor activates a small number of Kenyon cells across the mushroom body. These sparse and distributed activity patterns show minimal overlap, a feature that enables the mushroom body to represent many different odors. This feature is crucial for the memory function of the mushroom body: Kenyon cells converge onto a relatively small number of output neurons and these neurons are thought to instruct behavior. The hypothesis currently favored by the field is that, during learning, the synapse between the Kenyon cells and the mushroom body output neurons is modified. Thus, experience transforms a given odor representation into a learned behavior.

In this manuscript, Ocker and Buice describe a mathematical model of the Drosophila mushroom body that they use to explore how maximum performance is achieved under biological constraints. Namely, they are testing how the degree of distribution of connectivity and synaptic weight affect the performance of the model. Their findings suggest that flexibility in connectivity in combination with a fixed number of synaptic weights describes well the seemingly random connectivity patterns observed experimentally between projection neurons and Kenyon cells.

I am not sure what to make of this manuscript. I do like the question, as far as I understand it, and I do like that it seems to suggests a possible mechanism for setting up Kenyon cell connectivity during development. Because of my limited expertise in computational biology, I can only vouch for the biology used to build the model: it is sound and the assumptions made by the authors reflect our current understanding of the system. I will however have to defer to the editors and other reviewers to judge the quality of the work and its suitability for PLoS Computational Biology. I have a few major points about the writing of this paper, which I found incredibly cryptic.

Major points:

Major point 1: The biological basis of the model.

It is difficult for the reader to relate the computational work with the biology when reading this manuscript. As it stands, the biology stands alone and it is never referred to by the authors when introducing difficult mathematical concepts. For example:

Line 25-30: In simple neural network models, constraints define a space of possible synaptic weights (the "constraint space"), while the synaptic weights that perform a particular computation define another space (the "computational space"). The intersection of constraint and computational spaces defines the allowed synaptic weight (the "solution space").

While this (kind of) makes sense, it would be useful to remind the reader of the mushroom body analogy: what synapses and which computation are considered here? Another example is "flexibility". Are the authors referring to the flexibility of the projection neuron/Kenyon cell synapse or the Kenyon cell/output neuron synapse? Or some other flexibility? What do they mean? We need to know.

Major point 2: The mathematical jargon needs to be introduced and difficult concepts need to be digested.

This manuscript is difficult to read for non-mathematicians. The authors are often vague and use a lot of technical terms without clearly defining them. This might not bother the specialized audience of PLoS Computational Biology, but the paper would be more appealing and would reach a broader audience if it was better written. I am listing a few examples of concepts that are unbeknownst to me and some sentences that are especially cryptic:

Line 199: We computed the Laplace approximation to these model evidences.

Line 228: The Jeffreys prior is invariant under monotone changes of variables, resolving this paradox.

Moreover, it is especially important for the authors to explain their findings better rather than just referring to a figure and leaving the reader to their own device. For example:

Line 209: In the larva, the simplex area model outperformed the binomial model for both young and clawed KCs (Figure 3C).

What do the authors mean by "outperformed"? How can I reach this conclusion by looking at Figure 3C (a bar graph)?

There are many more such examples. I would recommend that the authors recruit a biologist on their team to try and make this manuscript readable for both mathematicians and biologists.

Major point 3: Consistency in the biological data.

While the biology used in this manuscript is sound, it is not clear to me why the authors using different systems, namely the larval and the adult mushroom body. Sometimes, the biology used to support their assumptions is yet borrowed from another system, hippocampal neurons for example. This lack of consistency makes the paper extremely confusing. Why not exploit one system, the larval mushroom body seems more appropriate here, and explain that system well and explicitly make the links between the biological system and the mathematical model.

Reviewer #2: Dear editor,

Thank you for the opportunity of reviewing the manuscript titled “Flexible neural connectivity under constraints on total connection strength.” In this manuscript, the authors derive a model for degree distributions of synaptic connectivity of a neuron under two different constraints: a) the bounded total synaptic strength, and 2) fixed total synaptic strength. They then compare the theoretical results against EM measurements of synaptic connectivity of KC’s in the mushroom body of Drosophila during different stages of development, and conclude that the bounded weight model is superior is explaining the synaptic connectivity of the Kenyon Cells (KC’s), particularly in the larval stage.

While the promise of the manuscript is interesting (namely that the neurons maximize their computational flexibility given certain constraints), I have some reservations in connecting their results to the promise, as I try to explain in what follows.

Major comments:

1) The authors initially make a clear case for the intersection between the constraint space and the solution space, and posit that he number of configurations of synaptic unites (vesicles or postsynaptic receptors) corresponding to the number of synapses is a reasonable marker of computational flexibility under given constraints. They even provide toy examples of this notion (lines 64 to 72.) The reader then expects to see whether the number of synapses in biology is tuned to the utilize the calculated maximum number of configurations under the postulated constraints (bounded total weight vs. fixed total weight.) However, that is not what the authors show. What they show is whether the in- (and out-) degree distribution of KC’s follow the distribution under the constraints, without any consideration of the maximum number of configurations. In other words, the authors do a great job delineating the constraint space, but not for its intersection with solution space. Only at the end of the results section, they look into the mode of the degree distributions (the supposed maximum number of configurations given the in/out degree and constraint). There, they show that the mode has a linear relationship with the number of synapses (K) in both models, with only the intercept (-1/2 vs. 2 in equation 12 and 13, respectively) being different between the two models. If this is the case, they should have examined the intercept in their experimental data, e.g. with a linear regression model.

2) It is not clear why the parameter \\alpha is introduced to relate synaptic units to synaptic strengths. Based on the theory developed before the experimental tests, it is expected that they consider a parameters such as the total number of synaptic vesicles for each neuron (or the total magnitude of postsynaptic densities) as a their measure of total synaptic units (\\bar{J}) and test their models against this, (i.e. how these units are distributed among synapses? Are the number of synapses tuned in such a way that the number of possible configurations of these units are maximized? etc.) But that’s not the analysis provided in the manuscript. Instead, the total number of synapses S is assumed to be proportional to J with an unknown constant of proportionality \\alpha. In simple terms, the data is too coarse to be able to test the model. Maybe the authors have a cogent explanation for how the data is related to the model, but it is not obvious to me. Importantly, they test the two models against a binomial distribution, but there is no \\alpha in the binomial case. Is it a fair comparison? It is also somewhat surprising why the binomial model is so poor compared to Poisson model, because the latter is a continuous approximation of the former. Am I missing anything here? There is also no comparison between binomial and the two models in testing for optimally flexible connectivity (figure 5.)

3) The description of the experimental data is very confusing. The authors should appreciate that PLoS Computational Biology is not a neuroscience journal, and even in a neuroscience journal with a wide audience it is not the case that all readers are fully familiar with the insect olfactory system. A schematic figure of mushroom body, its parts and the location of KC neurons, and their input/output will be very helpful. From the results (lines 137-156) it is not clear whether the input from projection neurons is included in the analysis or not. The same notion as in line 149 is repeated in line 154 (“KCs can be morphologically classified by their age”, and “The age of adult KCs can also be classified morphologically.”) This section requires a clear rewrite to my opinion, with an explicit explanation of what is being measured, what is exactly accounting for the in-degree, what is out-degree, how many neurons are being studied, how much sub-sampling happens in the measurements? How sensitive the results are to sub-sampling? etc.

Minor comments:

1) It would be helpful if the authors explain how they obtain eq 7. For me it is clear, as the length of the norm vector of the simplex surface is J/sqrt(K) and hence moving in the direction of the norm vector by dJ/sqrt(K) is equivalent to changing J by dJ. But it helps with the clarity of the manuscript if it is explained it in the Methods.

2) The subtitle “Measuring spaces of allowed circuit configurations” in discussion is not clear at all. In fact the same topic in Methods is more clearly explained!

3) Similarly, the subtitle “Maximum entropy connectivity distributions” is not clear. What questions are being addressed here?

4) In Equation 21, the second sum runs from n_2 = n_1 + 1, not from n_2 = n_1

5) In equation 24, the left hand side should not have index i.

6) Please break equation 54 in two lines. It falls off the margins.

7) The details of derivation of the Laplace approximation is not necessary (although it doesn’t hurt either.) More interesting would be asymptotic approximations of all the results for large N and K. This will give an intuition as to how the two models are different.

**Have all data underlying the figures and results presented in the manuscript been provided?**

Reviewer #1: Yes

Reviewer #2: Yes

PLOS authors have the option to publish the peer review history of their article (what does this mean?). If published, this will include your full peer review and any attached files.

Reviewer #1: No

Reviewer #2: No

---

## [Decision Letter · Decision Letter 1]

19 Jun 2020

Dear Mr. Ocker,

We are pleased to inform you that your manuscript 'Flexible neural connectivity under constraints on total connection strength' has been provisionally accepted for publication in PLOS Computational Biology.

Best regards,

Lyle J. Graham

Deputy Editor

PLOS Computational Biology

Reviewer's Responses to Questions

**Comments to the Authors:**

Reviewer #1: The paper is acceptable for publication. All points were addressed in the revised manuscript.

**Have all data underlying the figures and results presented in the manuscript been provided?**

Reviewer #1: Yes

PLOS authors have the option to publish the peer review history of their article (what does this mean?). If published, this will include your full peer review and any attached files.

Reviewer #1: No

---

## [Editor Report · Acceptance letter]

28 Jul 2020

PCOMPBIOL-D-19-01240R1 

Flexible neural connectivity under constraints on total connection strength

Dear Dr Ocker,

I am pleased to inform you that your manuscript has been formally accepted for publication in PLOS Computational Biology. Your manuscript is now with our production department and you will be notified of the publication date in due course.

With kind regards,

Sarah Hammond
